# Effects of Incorporating Alkaline Hydrogen Peroxide Treated Sugarcane Fibre on The Physical Properties and Glycemic Potency of White Bread

**DOI:** 10.3390/foods12071460

**Published:** 2023-03-29

**Authors:** Fatin Natasha Binte Abdul Halim, Afsaneh Taheri, Zawanah Abdol Rahim Yassin, Kai Feng Chia, Kelvin Kim Tha Goh, Suk Meng Goh, Juan Du

**Affiliations:** 1Food, Chemical and Biotechnology Cluster, Singapore Institute of Technology, 10 Dover Drive, Singapore 138683, Singapore; 2School of Food & Advanced Technology, Massey University, Private Bag 11222, Palmerston North 4410, New Zealand

**Keywords:** alkaline hydrogen peroxide, bread, glycemic potency, in vitro digestion, sugarcane fibre

## Abstract

The consumption of dietary fibres can affect glycemic power and control diabetes. Sugarcane fibre (SCF) is known as insoluble dietary fibre, the properties of which can be affected by physical, chemical, and enzymatic treatments. In this study, alkaline hydrogen peroxide (AHP) treatments were conducted over time (0.5, 1, 3, and 5 h) at 12.6% (*w*/*v*) SCF and the effects on the physicochemical and structural properties of the SCF were evaluated. After making dough and bread with the SCF, with and without AHP treatments, the glycemic responses of the bread samples were evaluated. Shorter durations of AHP treatment (0.5 and 1 h) reduced lignin effectively (37.3 and 40.4%, respectively), whereas AHP treatment at 1 and 3 h duration was more effective in increasing particle sizes (50.9 and 50.1 μm, respectively). The sugar binding capacity, water holding capacity (from 2.98 to 3.86 g water/g SCF), and oil holding capacity (from 2.47 to 3.66 g oil/g SCF) increased in all AHP samples. Results from Fourier-transform infrared spectroscopy (FTIR) confirmed the polymorphism transition of cellulose (cellulose I to cellulose II). The morphology of SCF detected under scanning electron microscopy (SEM) indicated the conversion of the surface to a more porous, rough structure due to the AHP treatment. Adding SCF decreased dough extensibility but increased bread hardness and chewiness. All SCF-incorporated bread samples have reduced glycemic response. Incorporation of 1, 3, and 5 h AHP-treated SCF was effective in reducing the glycemic potency than 0.5 h AHP-treated SCF, but not significantly different from the untreated SCF. Overall, this study aims to valorize biomass as AHP is commonly applied to bagasse to produce value-added chemicals and fuels.

## 1. Introduction

Dietary fibres can significantly contribute to physiological processes and maintain human health. They can lower the incidence of chronic diseases, like type 2 diabetes and cardiovascular problems. Adding dietary fibres to foods can lower glycemic potency and contribute to the prevention of diabetes [1]. Since the occurrence of diabetes continues to increase globally, recent research has attempted to find new ways of preventing diabetes. Adding dietary fibres to food products, such as white bread, has sparked interest among researchers and developers in the food industries [2,3,4]. Fibres are divided into two categories: soluble and insoluble. Although research has proven the health effects of soluble fibres to be more pronounced than insoluble fibres, insoluble fibres such as lignocellulosic materials are more abundant and affordable [5]. Modification of lignocellulosic materials with diverse methods (enzymatic, physical, and chemical) have the potential to increase its applications [5].

Sugarcane is cultivated extensively in Brazil, China, India, and other tropical regions of the world. In 2017, 1.84 billion tons of sugarcane were produced worldwide [6]. Processing 1 ton of sugarcane can generate approximately 280 kg of sugarcane bagasse. Sugarcane fibre (SCF) is an insoluble dietary fibre that can be processed from sugarcane bagasse [7]. As a lignocellulosic material, SCF is the major byproduct of sugarcane processing and is generally regarded as waste material. Finding optimal methods to incorporate SCF into food products, such as bread, a staple food, can benefit consumers and increase the net profit of the sugarcane industry [8]. Despite the benefits, adding SCF to bread can cause a loss in product quality, a decrease in bread loaf volume, inferior sensory characteristics, and increased hardness and chewiness of bread [3,9]. A study using Alkaline Hydrogen Peroxide (AHP) treatment was found to enhance such features in SCF-loaded bread [10].

AHP treatment was commonly used to pretreat bagasse to produce value-added chemicals and fuels [11]. AHP treatments usually involve solubilizing lignin and exposing hydroxyl groups, thereby improving the hydration properties of lignocellulose compounds [12]. Subjecting SCF to alkaline treatment can significantly affect the morphology of SCF as the alkaline environment causes fibres to swell and the cellulose molecules to detach from each other, thereby disrupting the rigid SCF structure [13]. This results in higher water absorbing SCF with a more open internal structure, allowing alkali-treated SCF to hydrate more, resulting in swelled and soft particles [14,15]. A relevant study by Meng et al. [16] considered the effects of AHP on the physicochemical and structural properties of buckwheat straw insoluble fibres. The results showed that AHP treatment caused wrinkled surfaces, due to the disintegration of the lignocellulosic part. Furthermore, breaking the intermolecular hydrogen bonds of cellulose and hemicelluloses through AHP treatment enhanced the water holding capacity of fibres. AHP treatment degrades lignin and does not release chemical wastes into the environment, primarily because hydrogen peroxide can decompose into oxygen gas and water and is an environmentally friendly and food-grade reagent. Another advantage of using AHP is that the duration of treatment is usually short, requires low pressure, and can operate with a low concentration [11,17] Therefore, AHP SCFs have the potential to be used as functional food ingredients. In a study done by Sangnark and Noomhorm [18], the water and oil holding capacities of SCF increased after AHP treatment at 1% hydrogen peroxide for 12 h. However, the incorporation of AHP SCF into bread caused a decrease in dough expansion, an increase in the hardness of bread, and a decrease in loaf volume, which reduced product quality [18]. In comparison, we aim to treat SCF with AHP at a higher concentration and shorter time to improve scalability, process mass intensity (PMI), and efficiency. 

Given the available literature and the knowledge gaps, this study aimed to explore the physicochemical properties of SCF when treated with 7.7% AHP at different durations of exposure (0.5, 1, 3, 5 h). Ultimately, the effects of adding SCF were evaluated on the physical properties and glycemic response of wheat flour-based dough and white bread.

## 2. Materials and Methods

### 2.1. Materials and Reagents

Food grade H_2_O_2_ (35% dilution, Raw Essentials, Philippines) was purchased from www.rawessentials.com.sg (accessed on 17 January 2021). Amyloglucosidase (EC 3.2.1.3 from *A. niger*, Megazyme, E-AMGDF, Bray, Wicklow, Ireland; 3260 U/Ml) was purchased from Megazyme. Absolute ethanol, 37% HCl, and glacial acetic acid were purchased from MERCK. NaOH pellets, 95–97% H_2_SO_4_, 99% ethanol, and sodium bicarbonate were purchased from AIK MOH, SINGAPORE, SINGAPORE. D-(+)-Glucose, sodium dodecyl sulphate, phenol, pepsin (pepsin EC 3.4.23.1 from porcine gastric mucosa, P7000, Sigma-Aldrich, Burlington, MA, USA; USP ≥ 250 U/mg), pancreatin (EC 232-468-9 from porcine pancreas, P7545, Sigma-Aldrich, Saint Louis, MO, USA; USP × 8 specifications), invertase (EC 3.2.1.26 from *S. cerevisiae*, I4504, Sigma-Aldrich, Saint Louis, MO, USA; ≥300 U/mg), maleic acid, sodium azide, calcium chloride dihydrate, sodium acetate, potassium sodium tartrate, and 3,5-dinitrosalicylic acid were purchased from Sigma Aldrich, Burlington, MA, USA.

### 2.2. AHP Modification of SCF

AHP modification of SCF was carried out following a method modified from Zhang et al. [11]. First, 1400 mL 10.5% H_2_O_2_ was prepared, and pH was adjusted to 11.5 with 5 M NaOH. Then, 240 g of SCF was added to the solution and stirred at 4 different durations (0.5, 1, 3, and 5 h), with manual titration of 5 M NaOH to maintain pH at 11.5. The final concentration of H_2_O_2_ was 7.7% while SCF in H_2_O_2_ was 12.6% (*w*/*v*). Subsequently, the solution was neutralized with 6 M HCl until pH reached 7.0, covered, and left overnight. Samples were transferred into centrifuge tubes and centrifuged at 4000× *g* for 10 min. Supernatant and pellet were collected separately. The pellet was rinsed with 2000 mL of deionized water with stirring in a beaker overnight. Centrifuging and rinsing were subsequently repeated twice to ensure thorough washing. Samples were collected into centrifuge tubes and stored at −80 °C for 24 h, and subsequent freeze drying (VirTis Benchtop Pro, SP Scientific, Warminster, PA, USA) at −80 °C for at least 72 h under vacuum condition. The final product of AHP modification at all 4 different durations (0.5, 1, 3, and 5 h) was a white powder with a yield of above 90%. SCF treated for 0.5, 1, 3, and 5 h were labelled as 0.5 h AHP SCF, 1 h AHP SCF, 3 h AHP SCF, and 5 h AHP SCF, respectively, while the untreated SCF was labelled as untreated SCF.

Based on the AHP treatment described above, the PMI value was significantly reduced from 101–200 [18] to 18.4. Details in the PMI calculation are described in Appendix A.

### 2.3. Characterization of SCF

#### 2.3.1. Acid-Insoluble Lignin Content Determination

Acid-insoluble lignin content was determined following a method modified from Fang et al. [19]. In a 250 mL round-bottomed flask, 1 g of SCF ($m_{0}$) in 100 mL ethanol was stirred and heated at 100 °C under reflux for 4 h. The mixture was then cooled to 25 ± 2 °C. Cold 15% H_2_SO_4_ was slowly added, and the mixture was stirred vigorously for 2 h at 25 ± 2 °C. Then, 560 mL of deionized water was added, and the mixture was stirred and heated at 100 °C for 4 h. The mixture was cooled to 25 ± 2 °C and subsequently filtered under vacuum conditions. The residue was collected and washed until neutral pH before drying in an air oven at 105 °C until dried. Mass ($m_{1}$) of the residue was recorded, and the procedure was repeated twice [19].
(1)$$Acid-insoluble\;Lignin\;Content\left(\%\right)=\frac{m_{1}}{m_{0}}\times 100\%$$

#### 2.3.2. Sugar Binding Capacity (SBC)

SBC was determined following a method modified from Dubey et al. [20]. First, 0.1 g of SCF was dispersed in 2.5 mL deionized water. Subsequently, 5 mL 0.01 M glucose was added, and the mixture was subjected to shaking under incubated conditions at 200 rpm and 37 °C for 6 h (SB-12l, Benchmark Scientific, Sayreville, NJ, USA). The mixture was centrifuged at 2000× *g* for 15 min, and the supernatant was filtered through a 0.45 µm filter. Then, 2 mL of filtrate was collected and subsequently analyzed using the phenol-sulfuric assay. Absorbance of the mixtures was recorded at 490 nm (Cary 60 UV-Vis Spectrophotometer, Agilent Technologies, Santa Clara, CA, USA) [20]. To quantify the glucose content of SCF, absorbance values of the samples were compared against a glucose standard curve.

#### 2.3.3. Water Holding Capacity (WHC) and Oil Holding Capacity (OHC)

The following method is modified from Zhao et al. [21]. First, 1 g of SCF and 20 mL of deionized water or canola oil were added to a centrifuge tube and mixed thoroughly using a vortex mixer for 1 min at highest speed. The sample was left to sit for 2 h at 25 ± 2 °C. After which, the sample was centrifuged at 3000× *g* for 30 min at 25 °C for WHC or 1500× *g* for 10 min at 25 °C for OHC. Without disturbing the pellet, the supernatant was removed using a 3 mL plastic dropper. After the complete removal of excess water or oil, the centrifuge tube containing the sample was weighed. The procedure was repeated thrice. WHC or OHC of SCF was determined following the equation below:(2)$$WHC\;or\;OHC=\frac{wet\;weight-dry\;weight}{dry\;weight}$$

#### 2.3.4. Particle Size Measurement

Particle size was determined following a method modified from Feng et al. [22]. First, 0.1 g of SCF in 10 mL of 1% Sodium Dodecyl Sulphate (SDS) solution was homogenized using IKA Homogenizer (Ultra TURRAX) at 6000 rpm for 2 min [22]. The particle size of the sample was then determined using parameters listed in Appendix A by Mastersizer 3000 (Malvern Instruments Ltd., HydroMV, Worcestershire, UK) with laser light scattering and ultrasonication, using the HydroMV dispersion unit. Results obtained were analyzed with Mastersizer 3000 software using the Mie scattering model.

#### 2.3.5. Fourier Transform Infrared Spectroscopy (FTIR)

A small amount of freeze-dried SCF sample was analyzed using an ATR FT-IR spectrophotometer (Agilent Cary 630 FTIR, Agilent Technologies, Santa Clara, CA, USA). The spectrum wavelength was recorded from 600–4000 cm^−1^ at a resolution of 4 cm^−1^ with 64 scans [14].

#### 2.3.6. Scanning Electron Microscope (SEM)

A small amount of freeze-dried SCF sample was scattered onto a conductive adhesive tape and subsequently coated with platinum for 60 s at a flow rate of 20 mA. The sample was then viewed in the vacuumed stage in the microscopic chamber of a scanning electron microscope (JSM IT800, Jeol Asia, Tokyo, Japan) [14].

### 2.4. Development of Dough and Bread

#### 2.4.1. Hydration of Flour Using DoughLAB

DoughLAB (DoughLAB 2500, Perten Instruments, Stockholm, Sweden) was used to ensure sufficient volume of water was incorporated into each bread formulation. It uses a 300 g mixing bowl based on the standard AACC International Approved Methods 54-21.02. [23] method. Two different measurements were conducted, one for 300 g wheat flour and one for 300 g wheat flour with 15 g of untreated SCF (based on 5% of baker’s percentage). For each measurement, water was added into the mixing bowl via an automatic water dripping system, followed by mixing at 30 ± 0.1 °C and speed 63 rpm for 20 min, whereby a maximum torque of 500 ± 25 Farinograph units (FU) indicated optimal dough development, with Farinograph results indicated in Appendix A. The water absorption values were recorded, and each analysis was conducted in triplicate. 

#### 2.4.2. Bread Making Process

Bread samples were prepared using the straight dough method, AACC International Method 10-09.01. [24], and the finalized bread formulations, with the respective water absorption determined in the previous analysis, are shown in Table 1.

Wheat flour was stored in the refrigerator overnight before use to ensure that its temperature is kept below 10 °C before usage. The amount of water was divided into 20% and 80% parts at 25 ± 2 °C and chilled (4 °C) conditions, respectively.

Dry yeast was pre-dispersed and hydrated in the 20% room temperature (25 ± 2 °C) portion of water for 10 min before mixing. All dry ingredients were pre-mixed using a stand mixer (3.3 L Mixer 5, Kitchen Aid, Greenville, OH, USA) with a spiral dough hook at speed 2 for 1 min before the yeast mixture and chilled water was slowly added in a steady stream. After the addition of water, mixing was continued at speed 2 for 2 min before increasing the mixing speed to speed 4 for 9 min. Approximately 20 g of dough was kept aside after mixing for dough extensibility analysis.

After mixing, 2 portions of dough weighing approximately 195 g each were molded into a ball, covered in aluminum foil, and left to rest at 25 ± 2 °C for 5 min. Dough pieces were placed in an incubator at 30 °C and relative humidity of 85% for 60 min to proof. After 60 min, they were taken out, knocked, and shaped into loaf pans before undergoing a final proofing at the same conditions for another 60 min. Finally, loaf pans were baked in an oven (MC28H5015AS, Samsung, Seoul, South Korea) at 200 °C for 30 min. After baking, bread samples were allowed to rest at ambient temperature (25 ± 2 °C) for 2 to 2.5 h before further analysis.

### 2.5. Characterization of Dough and Bread

#### 2.5.1. Dough Extensibility

The extensibility of dough was measured using the texture analyzer (Stable Micro Systems, TA.Xtplus Texture Analyzer, Godalming, Surrey, UK) with a Kieffer rig attached [25]. The texture analyzer was first calibrated with a 5 kg load cell before analysis. Immediately after mixing the ingredients listed in Table 1, the dough was clamped into the oiled Teflon mold and placed into the refrigerator to set for 30 min before analysis using parameters listed in Appendix A. Subsequently, a thin spatula was used to remove the dough strip from the mold and carefully place it onto the sample plate. The sample plate was then placed onto the Kieffer rig, and the measurement was repeated 10 times per formulation. After each analysis, the software quantifies the resistance to extension and extensibility of the dough sample.

#### 2.5.2. Loaf Specific Volume and Maximum Height of Loaf

Loaf specific volume of the bread samples was measured with the use of a solid displacement method based on the standard AACC International Method 10-05.01. [26] (Measurement of Volume by Rapeseed Displacement) with modifications made. In this analysis, mustard seeds were used in replacement of rapeseed. Mustard seeds were placed in and levelled in a container, and their volume was measured using a 500 mL graduated measuring cylinder. The bread sample was weighed before being placed into the same container, filled, and levelled with mustard seeds. The excess mustard seeds were measured using a 500 mL graduated measuring cylinder to determine the volume of the bread sample. The analysis was measured in duplicates, and the specific volume of bread was determined following the equation below:(3)$$Specfic\;Volume\left({cm}^{3}/g\right)=\frac{volume\;of\;bread\;sample}{mass\;of\;bread\;sample}$$

The maximum height of the bread sample was measured using a digital vernier caliper, measuring from the middle point of the longitudinal side of the bread. The analysis was done in duplicates per formulation.

#### 2.5.3. Texture Profile Analysis of Bread Crumb

The texture profile of the bread was analyzed using the texture analyzer (TA.Xtplus Texture Analyzer, Stable Micro Systems, Godalming, Surrey, United Kingdom) with a 36R probe attached [25]. The texture analyzer was first calibrated with a 5 kg load cell before analysis. Bread samples were sliced into 25 mm thickness, surface area ranging from 4500–5000 mm^2^, and placed onto the stage cross-section flat down for analysis. The texture analyzer then compresses the bread twice to understand how it behaves during chewing. After each analysis, the software quantifies the hardness and chewiness of the sample, and the test parameters are listed in Appendix A. The analysis was repeated thrice for each bread formulation.

#### 2.5.4. Moisture Analysis and Water Activity of Bread Crumb

The moisture content of bread samples was determined using a halogen moisture analyzer (HE53, Mettler Toledo, Columbus, OH, USA). Bread samples were sealed in Ziplock bags and kept in an airtight container at 25 ± 2 °C, and moisture content analysis was conducted the day after baking (Day 1). Approximately 4 g of bread sample was placed onto the sample pan and analyzed at 105 °C under automatic operating conditions. The analysis was repeated twice for each bread formulation.

The water activity of bread samples was determined using the water activity meter (Novasina, LabMaster-aw neo, Switzerland) at 25 °C. Approximately 1 g of bread sample was placed into the sample cup and analyzed. The analysis was repeated twice for each bread formulation.

#### 2.5.5. In Vitro Determination of Glycemic Potency

The glycemic potency of bread samples was determined, with slight modifications, via in vitro study as described by SRV et al. [27] and Monro et al. [28]. Bread samples were sliced, sealed in Ziplock bags, and stored at −20 °C. Prior to analysis, bread samples were thawed in ambient conditions (25 ± 2 °C) for at least 1 h.

#### 2.5.6. In Vitro Digestion of Bread Samples

First, 5 g of bread, in dry weight based on moisture content, was added to 50 mL of deionized water and homogenized with an S18N-19G dispersing tool (T18 digital ULTRA-TURRAX^®^, IKA^®^-Werke GmbH & Co.KG, Breisgau, Germany) at 6000 rpm for 2 min in a 250 mL Duran bottle. 

The mixture was adjusted to pH 2.5 with 1 M HCl. Subsequently, 2 mL of 10% (*w*/*v*) pepsin dissolved in 0.05 M HCl was added, to stimulate the gastric phase. The mixture was then subjected to shaking under incubated conditions at 170 rpm and 37 °C for 30 min (SB-12l, Benchmark Scientific, Sayreville, NJ, USA). The mixture was neutralized with 4 mL 1 M sodium bicarbonate and 10 mL 0.1 M sodium maleate buffer (pH 6), to initiate the small intestinal phase. Then, 500 µL aliquot (T_0_) of the suspension was transferred into 2 mL of chilled absolute ethanol and mixed thoroughly using a vortex to stop the enzymatic reaction. Next, 200 µL amyloglucosidase and 2 mL 5% (*w*/*v*) pancreatin dissolved in 0.1 M sodium maleate buffer were added to the neutralized mixture in the Duran bottle. Following the addition, the volume of the mixture was adjusted to 110 mL with deionized water. Mixtures were subjected to the same shaking and incubated conditions of 170 rpm and 37 °C, where successive 500 µL aliquots were removed at time points of 10, 20, 30, 40, 60, 120, and 180 min and transferred into 2 mL chilled absolute ethanol. Inactivated aliquots were kept at 4 °C prior to the quantification of glucose release. In vitro digestion of each bread sample was performed in duplicates, with timed samples removed in duplicates.

#### 2.5.7. Quantification of Glucose Release

The release of reducing sugars during the digestion process was quantified using the dinitrosalicylic acid (DNS) colorimetric method [27,28]. Inactivated aliquots were centrifuged at 1000× *g* at 25 °C for 10 min. Then, 500 µL of the supernatant was transferred to 250 µL of enzyme solution containing 1% (*v*/*v*) amyloglucosidase (and 0.05% (*w*/*v*) invertase dissolved in 0.1 M sodium acetate buffer (pH 5.2). The mixture was incubated at 37 °C for 15 min (SB-12l, Benchmark Scientific, Sayreville, NJ, USA). Subsequently, 750 µL of DNS mixture (comprising of a 1:1:5 mixture of 0.5 mg/mL of glucose solution, 4 M NaOH, and DNS reagent respectively) was added, and the contents were heated in a water bath at 95 °C for 15 min. Samples were cooled in ice water and 4 mL of deionized water was added and thoroughly mixed. Absorbance of the mixtures was recorded at 530 nm (Cary 60, Agilent Technologies, Santa Clara, CA, USA). To quantify the glucose content of SCF, absorbance values of the samples were compared against a glucose standard curve. 

### 2.6. Data Analysis for In Vitro Glycemic Potency Determination

The release of glucose through in vitro digestion was quantified via the dinitrosalicylic acid (DNS) colorimetric method [27,28]. The glucose content of bread samples was derived from a glucose standard curve (y = 0.0965x + 0.2459) and subsequently used in the calculations of Glycemic Glucose Equivalent (GGE) values per 100 g of bread. The GGE release curves were depicted by plotting the GGE values against the in vitro digestion time. Equation (4) was used for deriving an apparent glucose disposal rate (GD). The GD values were subsequently used for deriving the GGE disposal per time point (GD × time point = GGE disposal), which was then plotted against the in vitro digestion time. Subsequently, to account for blood glucose clearance, the GGE disposal values were subtracted from the GGE release to generate net GGE values (Net GGE = GGE−GD). To account for the in vivo delay in the onset of the glycemic response, the in vitro digestion time was extended for an additional 10 min. These values were then plotted against the net GGE values. The incremental area under the curve (iAUC) was calculated based on the net GGE curve using the 45 triangle/trapezoid summation method described by the World Health Organization (WHO) [28], as presented in Appendix A. The relative glycemic potency of bread samples was determined on an equal weight basis of 100 g of bread in comparison with the iAUC of a white bread reference of known GGE. The glycemic index (GI) of bread samples was then estimated by comparing the calculated iAUC of bread samples with the iAUC of reference bread.
(4)Glucose Disposal (GD)=0.0135x+0.02232
where x = GGE released by 100 g of bread at 40 min of in vitro digestion.
(5)$${GI}_{bread\;sample}=\frac{iAUC\;of\;bread\;sample}{iAUC\;of\;reference}\times {GI}_{reference}$$

### 2.7. Statistical Analysis

Statistical analyses were performed with the one-way analysis of variance (ANOVA) test using Minitab software (Minitab^®^19, MiniTab Inc., State College, PA, USA). The means and standard deviation were compared using Tukey’s honest significance test at 95% confidence level, where a significant difference is observed when *p* < 0.05. Analytical values of experiments are shown as mean ± standard deviation.

## 3. Results and Discussion 

The acid-insoluble lignin content, sugar binding capacity (SBC), and particle size presented as D_(4,3)_ and D_(3,2)_ of the untreated SCF and AHP-treated SCF samples are listed in Table 2.

Accordingly, the acid-insoluble lignin content of SCF decreased after the AHP treatment at 0.5 h but increased again at 3 h (Table 2). Similar decreases in the lignin content of fibres were reported after applying AHP treatments to lignocellulosic materials in previous cases of relevant research [5,11,29]. Hydrogen peroxide is known as a strong oxidizing agent that can cause delignification and solubilization of hemicellulose. According to Gould [15], an alkaline medium (pH = 11.5) can hasten the decomposition of hydrogen peroxide. During the decomposition of hydrogen peroxide, hydroperoxide ions are released, thereby contributing to the removal of chromophoric groups of lignin by attacking carboxyl groups and ethylene groups. Hydrogen peroxide then reacts with hydroperoxide ions to generate hydroxyl radicals which, in turn, are regarded as the strongest oxidizers for having an important role in the oxidation process [15]. The delignification and solubilization of hemicellulose in sugarcane fibres usually result from the activity of hydroxyl radicals [30]. In our study, AHP treatment reduced the lignin content in all samples. The acid-insoluble lignin content decreased significantly in the 0.5 h and 1 h AHP SCF samples. However, as the AHP treatment time increased from 1 h to 5 h, a slight increase occurred in the acid-insoluble lignin content, which was noticeable when comparing the 0.5 h and 1 h AHP SCF samples to the 3 h and 5 h AHP SCF samples. Our results agree with another study, which showed that AHP treatments caused a significant decrease in lignin content in the first 30 min, although an increase in the treatment time caused no further declines in the lignin content [31]. However, the acid-insoluble lignin content increases to 65% and 65.7% for 3 h and 5 h, suggesting that hemicellulose has been reduced as indicated in the FTIR spectra. As seen in Table 2, there was an increase in the SBC of SCF samples treated with AHP. Fibres usually have the capacity to bind to glucose and, thus, can potentially lower postprandial blood glucose concentrations [32,33]. As shown in Table 2, particle sizes of AHP SCF samples were larger than those of untreated-SCF samples. The particle size of AHP SCF samples increased from 0.5 h to 1 h but then decreased slightly as the treatment time increased to 5 h. This agrees with a study done by Gu et al. [34], which indicated that reduction in particle size was due to the delignification effect of AHP on corn stover and wheat straw. Additionally, this correlated with its WHC since a positive correlation usually exists between the particle size and the WHC of SCF [35]. As the WHC increased, there is an increase in hydrodynamic diameter, which is indicated by D_(3,2)_ value due to its swelling capacity and, thus, led to a greater amount of water absorption.

Figure 1 shows the WHC and OHC of both AHP and untreated-SCF samples. The WHC values for untreated SCF, 0.5 h AHP SCF, 1 h AHP SCF, 3 h AHP SCF, and 5 h AHP SCF were 2.98, 3.18, 3.86, 3.59, 3.38 g of water per g of SCF, respectively. As compared to the untreated SCF, the WHC of SCF increased after the AHP treatment, rising through time, from 0.5 to 1 h, but then decreased thereafter until the fifth hour. An increase in WHC values reportedly occurred in previous research on ginseng fibre [29], okra fibre [5], and sugarcane fibre [18,36]. The increase in the WHC of AHP-treated SCF could be due to the changes in morphological structure of the fibres via the breakage of bonds and a greater exposure of hydroxyl groups [29,37].

The OHC values for untreated SCF, 0.5 h AHP SCF, 1 h AHP SCF, 3 h AHP SCF, and 5 h AHP SCF were 2.47, 3.66, 3.50, 3.18, and 2.96 g of oil per g of SCF, respectively. As compared to untreated SCF, the OHC of SCF increased after the AHP treatment. While the OHC values of AHP SCF samples were higher than that of the untreated SCF, it decreased over time, from 0.5 h to 5 h. Similar studies reported an increase in OHC after subjecting SCF to AHP treatments [29,38]. Additionally, the observed trend in OHC values could be attributed to the increased porosity of SCF after the AHP treatment.

FTIR analysis was conducted to identify the functional organic groups in SCF and to monitor their alterations before and after AHP treatments. Figure 2a depicts the FTIR spectra of SCF samples, where absorbance peaks appeared at 3200–3500 cm^−1^ (OH groups in cellulose), 2892 cm^−1^ (C-H vibration in cellulose and hemicellulose), 1633 cm^−1^ (vibration of water molecules absorb in cellulose), 1426 cm^−1^ (–CH_2_ group of cellulose), 1367 cm^−1^ (–CH group of cellulose), 1158 cm^−1^ (hydroxyl bending of cellulose), 1028 cm^−1^ (O–H vibration in cellulose), and 895 cm^−1^ (C–O stretching in cellulose). The broadening of the OH band (3200–3500 cm^−1^) in IR spectra resulted from a mixture of intermolecular (3230–3310 cm^−1^) and intramolecular (3340–3375 cm^−1^) hydrogen bonds [39,40]. These bands have higher intensities in the spectra of AHP-treated SCF, thereby confirming the variation in the hydrogen bond structure of SCF. The –CH region (2870–2900 cm^−1^) was used as an indicator of polymorphic transformation within the crystalline parts of cellulose [41] In this study, after 0.5 and 5 AHP treatments, the peak values of the CH region shifted from 2884 to 2890 and 2893 cm^−1^ (Figure 2b), respectively, which indicated the polymorphism transition from cellulose I to cellulose II [42]. The peak at 1633 cm^−1^ emanated from absorbed water molecules via hydrogen bonds in the amorphous region of cellulose macromolecules. AHP caused a higher intensity of the peak (1633 cm^−1^), which amplified the decrease in the amorphous part of the SCF during the AHP treatments (Figure 2c). Some specific peaks at 1426 and 895 cm^−1^ were relevant to the spectrum of the treated SCF, which relate to the CH bending of amorphous and crystalline cellulose I. The empirical crystallinity index (CI) can be calculated using the ratios of peaks at 1426 and 895 cm^−1^ [43]. The AHP treatment increased the CI index (from 0.452 of untreated SCF to 0.490, 0.491, 0.499, and 0.525 of 0.5 h to 5 h AHP SCF), revealing that the samples mainly comprised crystalline cellulose II. The peak at 1277 cm^−1^ indicated C-H bending and O-H bending in plain deformation, representing an overwhelming existence of crystalline cellulose II (Figure 2c). As the absorbance at 895 increased with 0.5 h AHP SCF, showing the transformation from cellulose I (natural cellulose) to cellulose II (because cellulose II is more stable than cellulose I), the results confirmed the polymorphism transformation peaks, shifting at 2800 cm^−1^.

As seen in Figure 3a,b, the untreated-SCF samples appeared larger lengthwise and had a smoother surface, probably due to the presence of intact lignin and hemicellulose. In comparison, AHP SCF samples were more fragmented and had a rougher, wrinkled, more porous surface (as indicated by the yellow arrows). Subjecting SCF to sodium hydroxide can significantly affect the morphology of SCF as the alkaline environment causes fibres to swell and the cellulose molecules to detach from each other, thereby disrupting the rigid SCF structure [13].

AHP treatment could have caused shorter fragments and rough surfaces in SCF (Figure 3b–e and Figure 3g–j). With extended hours of AHP treatment, FTIR results indicated that amorphous cellulose, lignin, and hemicellulose content decreased from 0.5 to 5 h, which caused the SCF to be more porous, fragmented, and wrinkled, and it agrees with the SEM results. In a relevant study, AHP treatment with the effect of reducing hemicellulose in the fibres caused a rougher surface in kenaf fibre [44]. The non-uniformity of fibre surface, increasing porosity, and wrinkle structure were observed in citrus fruit fibre [38], white turnip fibre [45], and ginseng fibre [29]. The wrinkled, porous surface of the AHP-treated SCF samples meant a larger surface area and a greater level of exposure to hydrophilic groups, thereby increasing the WHC, as discussed previously [29,46]. The particle size of SCF increased and then decreased from 0.5 to 5 h as indicated in Table 2, which suggests the size of SCF swelled first and then dissembled, from Figure 3a–e.

In Table 3, the reference sample refers to a typical white bread sample with no SCF incorporation, while the untreated-SCF bread sample refers to that incorporated with untreated SCF. Bread samples of 0.5 h AHP SCF, 1 h AHP SCF, 3 h AHP SCF, and 5 h AHP SCF refer to samples with the corresponding AHP SCF samples. Table 3 lists the resistance to extension of dough samples collected after mixing, maximum height, and specific volume of bread samples.

According to Table 3, a greater level of resistance to the extension of dough samples was observed in the untreated group and the 0.5 h AHP SCF incorporated bread, compared to that of the reference sample. Even though 1 h and 5 h AHP SCF dough samples had lower resistance to extension values compared to the reference, the samples were observed to break off immediately during the analysis, when stretched by the texture analyzer. This observation was in agreement with a previous study on the incorporation of wheat bran dietary fibre on steamed bread, showing that the incorporation of fibre resulted in a decrease in dough extensibility [47]. Another study conducted by Gomez et al. [48] suggested that the incorporation of dietary fibres increased the resistance to extension but decreased the dough extensibility. The decrease in dough extensibility and the increase in stiffness of the dough resulted from disruptions in the gluten network structure and reduced gas retention ability through the process of dough and bread making because SCF was incorporated [48]. This could negatively affect the height and loaf volume of bread in the production line [47,49]. As shown in Table 3, the maximum height of AHP SCF-incorporated bread samples significantly decreased, compared to the reference sample.

The decrease in maximum height can be seen in Figure 4a. Similar studies on the incorporation of SCF [10] and hazelnut fibre [50] showed a decrease in height and loaf volume, most probably due to lower gas retention during proofing resulting from the disruption of the gluten network. Figure 4b demonstrates the average hardness and chewiness of bread samples. The average hardness values for the reference sample, control sample, 0.5 h, 1 h, 3 h, and 5 h AHP SCF were 2398, 2892, 4058, 4185, 3878, and 3794 g, respectively; while the average chewiness values were 2187, 2218, 2604, 2960, 2444, and 2567. By adding SCF, the average hardness and chewiness increased compared to the reference sample. In addition, all AHP samples are significantly harder and chewier than the untreated-SCF sample, but no significant difference was found among all AHP samples. Previous studies done by was found that incorporating dietary fibre from wheat bran and inulin [51] and fenugreek [9] have increased the hardness of bread. The hardness and chewiness trends correspond to the WHC values of SCF samples due to the amount of water in the formulation of all SCF-incorporated bread samples, which were kept constant despite having an increase in WHC. The increase in the hardness of AHP SCF incorporated bread samples was due to the SCF, which competed with gluten over the absorption of water, as the AHP SCF had a higher WHC and therefore resulted in a weak gluten network, which was unable to retain large gas bubbles produced during proofing [52].

Figure 5 shows the respective glycemic glucose equivalent (GGE) disposal, GGE release, and net GGE curves of the bread samples. GI values are categorized as low (≤55), medium (55 < GI < 70), and high (≥70) [53]. White bread had a high GI value of 70–100, which can be used as a reference to calculate the estimated GI values of SCF-incorporated bread samples. In this study, the reference sample was taken to have an estimated GI of 100.

Table 4 lists the incremental area under the curve (iAUC), relative glycemic potency (RGP), and rapid, digestible starch (RDS) values of the respective bread samples. There was a significant decrease in iAUC values, associated with the incorporation of SCF, compared to the reference sample. A similar trend was also observed in both RGP and RDS values of the SCF-incorporated bread samples. Additionally, the 0.5 h AHP SCF incorporated sample has significantly higher iAUC, RCP, and RDS values than untreated SCF and the rest of the AHP SCFs, but there is no significant difference found among untreated SCF and AHP SCF samples. As shown in Table 4, the estimated GI values of SCF-incorporated samples were lower than that of the reference, making them fall under the “medium” or “low” GI category. However, there was a slight increase in the respective values of the 0.5 h AHP SCF samples, compared to the other AHP SCF samples. This could suggest that a short duration of AHP treatment at 0.5 h increased the digestion rate, but a longer duration of the AHP treatment (1 h and above) decelerates the digestion rate. In a study by Dhital et al. [54], starch hydrolysis can be inhibited by a high concentration of cellulose during digestion. Cellulose can bind to α-amylase via a noncompetitive mechanism, which can effectively inhibit its activity on starch. A similar study by Seki et al. [55] demonstrated that postprandial blood glucose concentrations were lowered by insoluble dietary fibre. We did observe that the 0.5 h AHP SCF sample has a lower glycemic response reduction effect. This suggests that SCF treated at shorter AHP durations were unable to modulate the degree of starch gelatinization. However, at 0.5 h AHP treatment, lignin was removed, indicating that hemicellulose reduction might be able to further influence starch digestibility as further reduction of hemicellulose would cause a disintegrated fibre structure, therefore increasing surface area. Nsor-Atindana et al. [56] demonstrated that particle size and surface area influenced the interaction of cellulose with α-amylase and as a result, the digestion of potato starch in both oral and gastric digestion, with the inhabitation effect of cellulose against α-amylase becoming more pronounced as particle size was reduced while surface area increased. Although in our study, significant reduction of GI was not observed in the AHP SCF incorporated breads in comparison to the untreated one, which could be limited by the low amount of SCF incorporated (5%) or a higher amount of water added as compared to the reference, which was determined by farinograph analysis. In future studies, we suggest incorporating dough improvers, such as ascorbic acid, to reduce the dilution and destruction effect of SCF in the gluten network while incorporating a higher amount of SCF to amplify its effect on reducing starch digestibility. However, this study offers guides for future researchers to use optimized AHP duration (3 h) with improved PMI to treat SCF, to increase its WHC while not to impact the starch digestion reduction ability of SCF.

## 4. Conclusions

To improve the physicochemical properties of SCF fibre, a high hydrogen peroxide concentration (7.7%) and shorter time (0.5, 1, 3, and 5 h) were used in this study to increase the productivity and scalability of the alkaline hydrogen peroxide (AHP) treatment. The AHP treatment affected the composition, structure, and physicochemical properties of SCF. The acid-insoluble lignin content of the SCF decreased in response to the AHP treatment from 0.5 h to 5 h under the condition of 7.7% H_2_O_2_ at pH 11.5. With the AHP treatment, the SCF exhibited higher WHC and OHC values, compared to the untreated SCF. This could result from the exposure of hydrophilic groups and cause an increase in the surface area as well as the porosity of SCF. The greater surface area and porosity of the SCF after AHP treatment were found and apparent in the SEM images. Although the AHP treatment positively affected the physiochemical properties, its incorporation into wheat flour-based dough and white bread had a negative effect on textural properties. Dough and bread samples incorporated with AHP SCF exhibited a reduction in dough extensibility and maximum height, associated with an increase in the hardness and chewiness of bread. This resulted from the disruption in the gluten network structure and reduced gas retention ability in the process of dough and bread making, as the incorporation of SCF resulted in competition with gluten for the absorption of water, especially the AHP SCF which had higher WHC. Despite its impact on textural properties, the incorporation of SCF successfully lowered the estimated GI values of bread. Incorporation of 1, 3, and 5 h AHP SCF was effective in reducing the glycemic index than 0.5 h AHP SCF. We need further studies to explain the mechanism of AHP treatment on the glucose release rate of SCF-incorporated bread.

## Figures and Tables

**Figure 1 foods-12-01460-f001:**
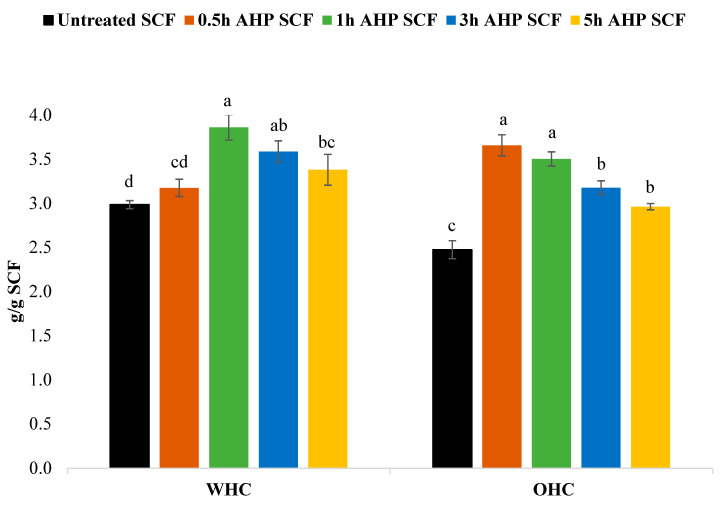
WHC and OHC of untreated and AHP SCF, different letters represent a significant difference within the same functional test (*p* < 0.05), where *n* = 3.

**Figure 2 foods-12-01460-f002:**
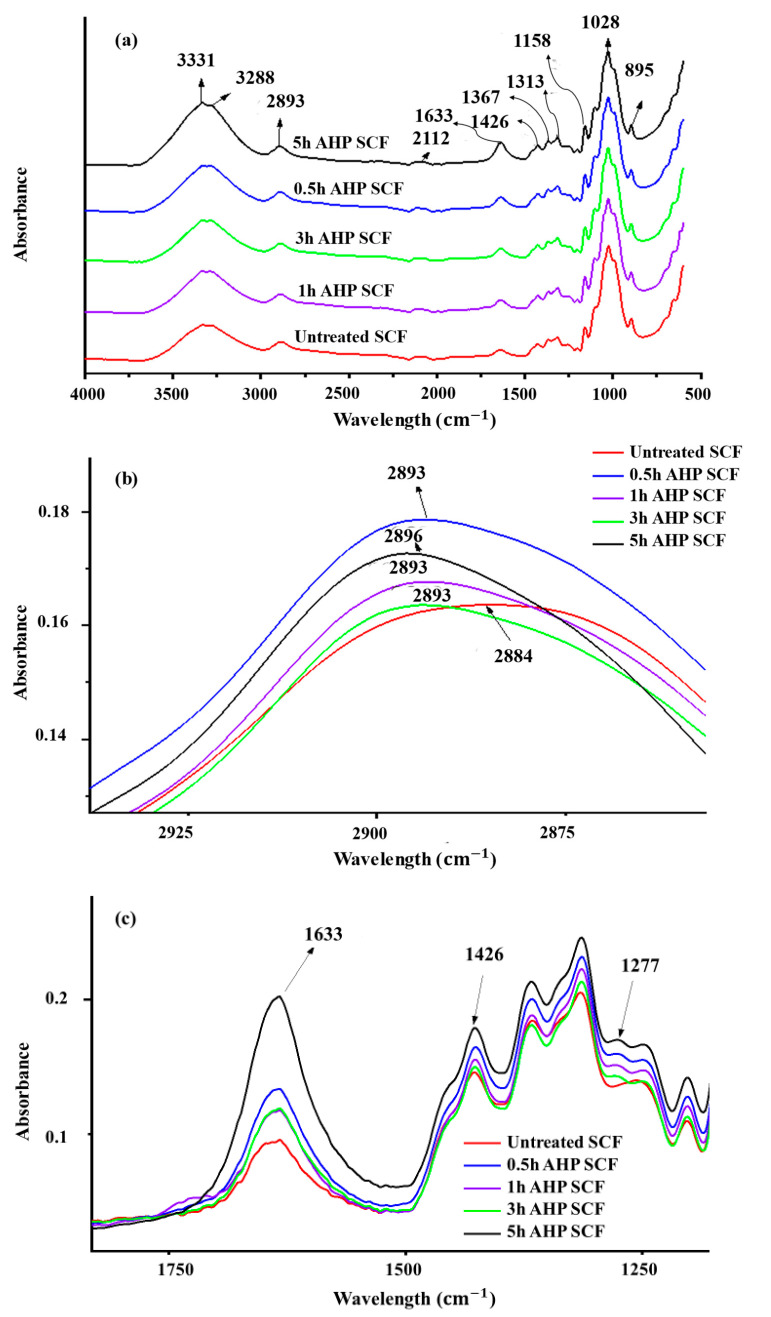
FTIR spectra of untreated SCF, 0.5 h, 1 h, 3 h, and 5 h AHP SCF (**a**) at the wavelength of 500–4000 cm^−1^, (**b**) The –CH region (2870–2900 cm^−1^) shifting, and (**c**) Variations of peak intensity at wavelength 1633 (cm^−1^) and indicator peaks in transformation cellulose I to cellulose II (1426 and 1277 cm^−1^).

**Figure 3 foods-12-01460-f003:**
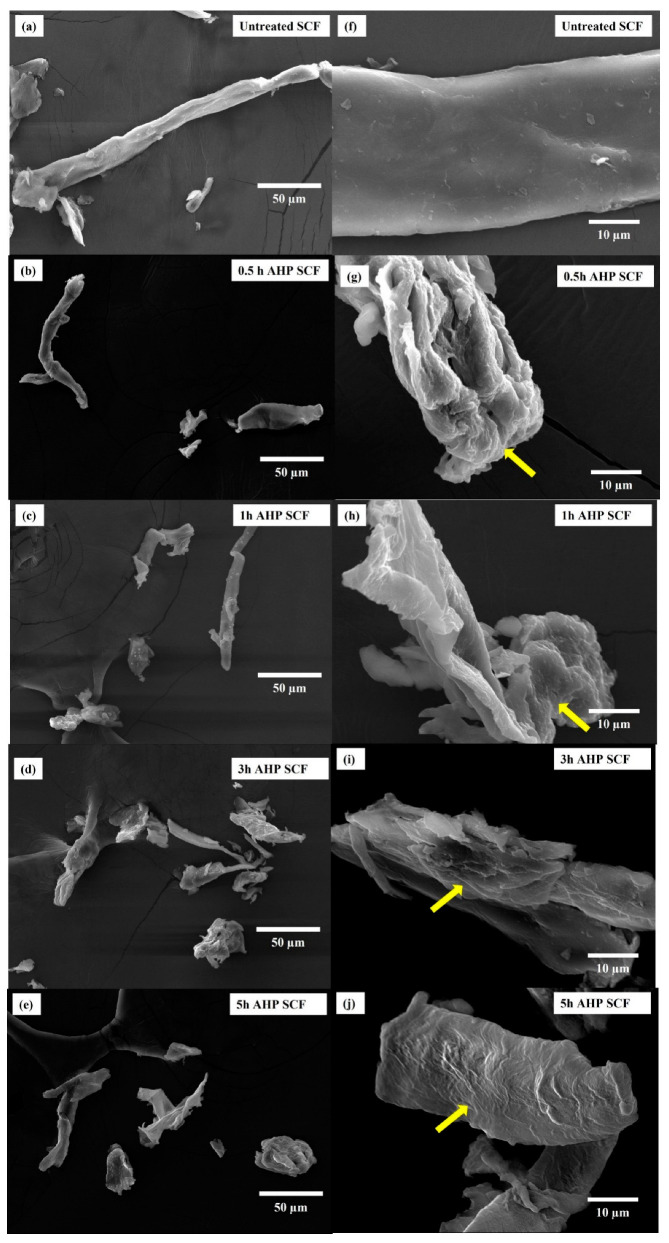
Morphological images of untreated and AHP SCF detected by SEM at (**a**–**e**) 500× magnification and (**f**–**j**) 2000× magnification. Yellow arrows indicate surface differences.

**Figure 4 foods-12-01460-f004:**
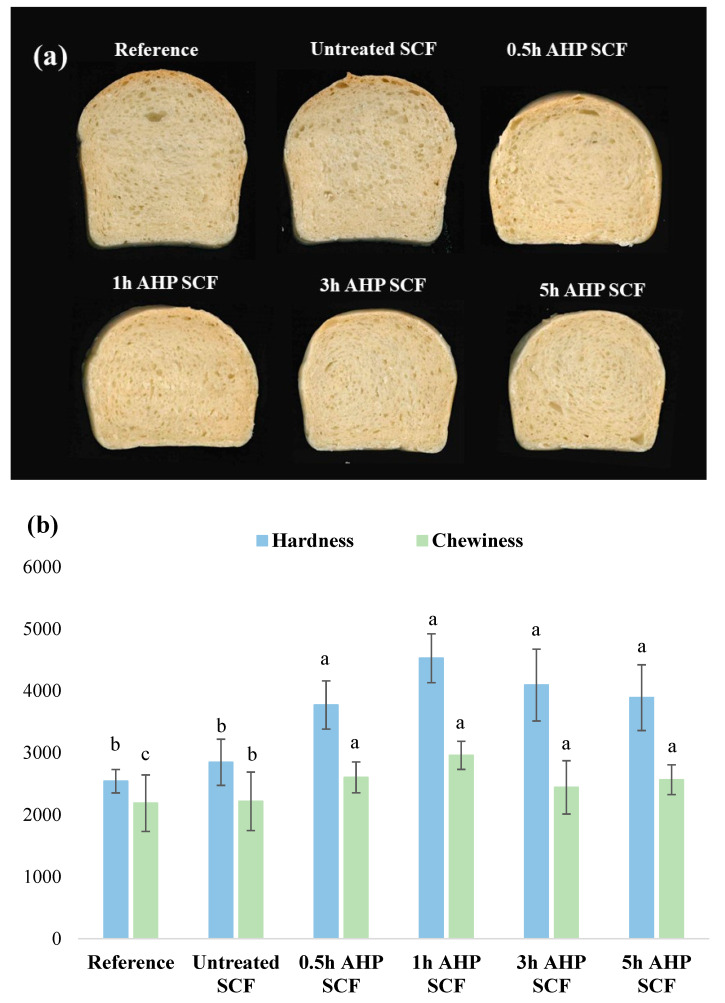
(**a**) Cross-sectional photographs illustrating the crumb appearance of bread samples. (**b**) Texture profile analysis (hardness and chewiness) of bread samples without SCF (reference), incorporated with 5% untreated SCF, and 0.5 to 5 h AHP SCF. Different letter represents a significant difference within the same texture attributes (*p* < 0.05), where *n* = 6.

**Figure 5 foods-12-01460-f005:**
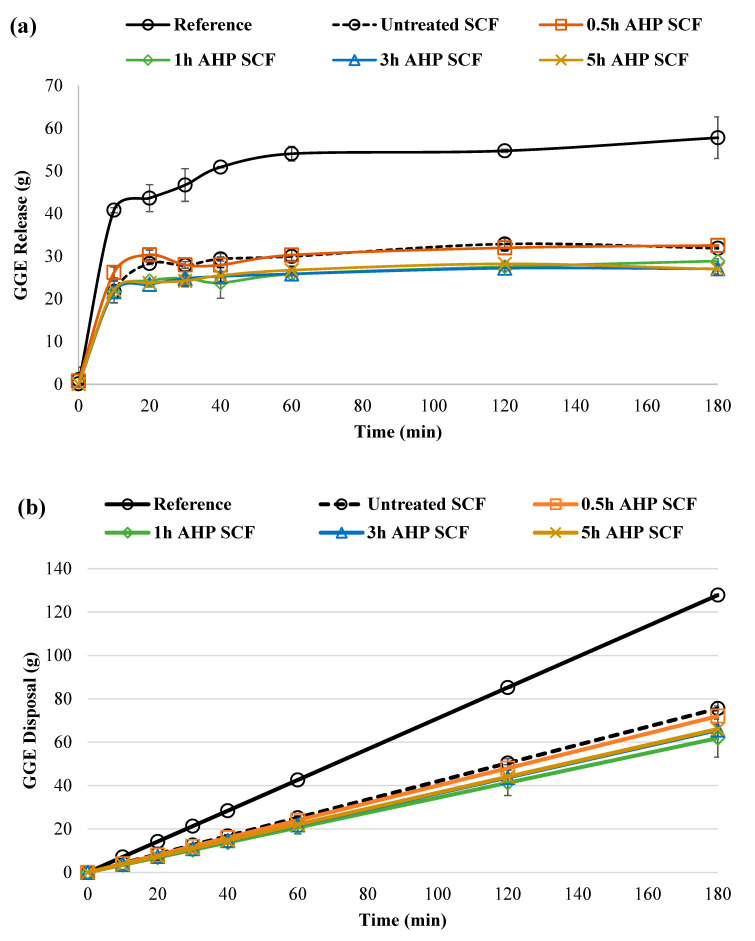
(**a**) GGE release curves illustrating overall glucose release. (**b**) GGE disposal curves. (**c**) Net GGE curves. Error bars represent mean ± standard deviation (*n* = 2). All data were calculated based on the fresh weight of the bread samples without SCF incorporated (reference), incorporated with 5% untreated SCF, and 0.5 to 5 h AHP SCF.

**Table 1 foods-12-01460-t001:** Formulation of reference and SCF incorporated breads.

	Reference		SCF	
Ingredients	Mass (g)	*w*/*w* (%)	Mass (g)	*w*/*w* (%)
Wheat Flour	300.0	59.6	300.0	56.2
SCF	-	-	15.0	2.8
Water	193.5	38.5	208.8	39.2
Dry Yeast	3.60	0.7	3.60	0.7
Salt	6.00	1.2	6.00	1.1
Total	503.1	100.0	533.4	100.0

**Table 2 foods-12-01460-t002:** Acid-insoluble lignin content, sugar binding capacity (SBC), particle size of untreated SCF and AHP SCF samples.

Sample	Acid-Insoluble Lignin (%)	SBC (g/g SCF)	Particle Size (µm)	
			D_(4,3)_	D_(3,2)_
Untreated SCF	88.7 ± 0.15 ^a^	3.8 × 10^−5^ ± 2.3 × 10^−6 b^	46.1 ± 0.47 ^d^	27.8 ± 0.48 ^c^
0.5 h AHP SCF	55.6 ± 0.29 ^c^	4.2 × 10^−5^ ± 3.0 × 10^−7 a^	49.3 ± 0.62 ^c^	28.0 ± 0.18 ^c^
1 h AHP SCF	52.8 ± 0.32 ^c^	4.3 × 10^−5^ ± 1.7 × 10^−6 a^	50.9 ± 0.50 ^a^	29.9 ± 0.12 ^a^
3 h AHP SCF	65.0 ± 0.70 ^b^	4.1 × 10^−5^ ± 3.7 × 10^−8 ab^	50.1 ± 0.72 ^b^	29.9 ± 0.11 ^a^
5 h AHP SCF	65.7 ± 4.03 ^b^	4.2 × 10^−7^ ± 2.5 × 10^−7 a^	49.7 ± 0.37 ^bc^	29.4 ± 0.25 ^b^

Different letter represents a significant difference (*p* < 0.05) in the same column, where *n* = 3.

**Table 3 foods-12-01460-t003:** Extensibility of dough samples, maximum height, and specific volume of bread samples without SCF incorporated (reference), incorporated with 5% untreated SCF, and 0.5 to 5 h AHP SCF.

Samples	Resistance to Extension (g)	Maximum Height (mm)	Specific Volume (cm^3^/g)
Reference	20.08 ± 0.45 ^c^	61.7 ± 1.9 ^a^	2.24 ± 0.05 ^a^
Untreated SCF	26.62 ± 1.87 ^b^	61.6 ± 1.5 ^a^	2.20 ± 0.05 ^ab^
0.5 h AHP SCF	30.41 ± 1.04 ^a^	57.6 ± 3.2 ^b^	2.13 ± 0.04 ^ab^
1 h AHP SCF	13.10 ± 1.05 ^d^	54.1 ± 0.7 ^b^	2.00 ± 0.15 ^b^
3 h AHP SCF	20.56 ± 2.71 ^c^	54.2 ± 0.4 ^b^	2.14 ±0.09 ^ab^
5 h AHP SCF	11.72 ± 0.88 ^d^	54.7 ± 0.9 ^b^	2.05 ± 0.18 ^ab^

Different letter represents a significant difference (*p* < 0.05) in the same column, where *n* = 6.

**Table 4 foods-12-01460-t004:** Estimated glycemic index (GI) values of bread samples with no SCF incorporated (reference), incorporated with 5% untreated SCF, and 0.5 to 5 h AHP SCF based on fresh weight basis.

Samples	Estimated GI		
	iAUC	RGP	RDS
Reference	100	100	100
Untreated SCF	56	60	65
0.5 h AHP SCF	63	58	70
1 h AHP SCF	53	50	56
3 h AHP SCF	49	50	54
5 h AHP SCF	51	52	54

## Data Availability

Data is contained within the article.

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
