# Peer review of "Effects of Incorporating Alkaline Hydrogen Peroxide Treated Sugarcane Fibre on The Physical Properties and Glycemic Potency of White Bread"

_foods, 2023, doi:10.3390/foods12071460_

Round 1

Reviewer 2 Report

This manuscript investigated the effects of incorporating alkaline hydrogen peroxide (AHP) treated sugarcane fiber on the physical properties and glycemic potency of white bread. AHP treatment is commonly used to pretreat fibers to change their physicochemical and functional properties, while the manuscript still has some important points, and the results are readable. It may be of potential interest to the reader. However, there are some flaws in the manuscript that affect the significance of the reported results.

1.     Line 293, what does “SBC” mean here?

2.     In Table 2, “0.5H AHP SCF”, “1H AHP SCF” and “3H AHP SCF” , what does “H” mean after 0.5,1,3 ? I think it means hour, but I am not sure it should be “H” or “h”.

3.     Why 7.73% AHP? Did the author investigate different concentrations of AHP?

4.     The fiber was pretreated at pH11.5, and then added into the bread, will the pH11.5 affect the taste or the safety of the bread? And why 12.6 (w/v) SCF?

Reviewer 3 Report

The paper ''Effects of Incorporating Alkaline Hydrogen Peroxide Treated  Sugarcane Fibre on The Physical Properties and Glycemic Potency of White Bread'' is a research work that deals with flour replacement by sugarcane fibre treated with hydrogen peroxida at different time intervals. The investigation of the structural changes of fibre and the measurement of the physical properties and glycamic potency of the final breads is interesting and a lot of information is combined. The paper has the drawback that the doughs with different fiblre treatments contain the same water content and consequently the firmness of the final breads increases. I  would suggest to change the water content and thereafter make any comparisons among breads.

Further changes are suggested:

Lines 320-323 clarify further why there is a decrease in particle size and then an increase in it. 

lins 338-340 Are these statements visible in SEM photos?

Fig. 1 indicate the columns between which comparisons were made 

Fig. 4b Please indicate the columns inbetween comparisons were made.

Fig. 2c I would suggest to show the second derivative rather the original courves ,Maybe clearer information is evident.

417-432 I would suggest the authors add experiments at different water content according to the water holding capacity of the composite flours and thereafter compare SV and textural parameters (see general comment as well)

Fig.5 Water content does not influence the results taken? Tihs means by changing the water amount  in the doughs results would be similar? Please add some comments. (same as previously)

Reviewer 4 Report

Manuscript number: foods-2239611 entitled "Effects of Incorporating Alkaline Hydrogen Peroxide Treated Sugarcane Fibre on The Physical Properties and Glycemic Potency of White Bread ".

Some weak points must be revised and checked. Several references were too old. Specific comments are:

Abstract

- The significant numerical data must be added.

- L. 11, it should be revised as “glycemic response”.

Introduction

- Please check reference such as L.66.

- L.78, you mentioned about 1% hydrogen peroxide for 12 h and you would like to reduce time for improving scalability and efficiency. Thus, the sample of SCF treated with this condition should be used as control but you used only 0.5, 1, 3 and 5 h.

Materials and methods

- L. 91, A. niger must be italic

- Again, L.103 check reference.

- L. 108, it should be used as g force not rpm.

- L. 172, why did you use 15 g of SCF? How come?

- L. 186, all room temperature should be avoided. Please specify.

- L. 200, you held the sample at least 2 h but it should be included the maximum time also. It affects to some properties that you measured, for example, texture.

- GI calculation should be explained more in methodology.

Results and discussion

- Table 3, the specific volume seemed not to be related to Fig. 4a. The sample 5H AHP SCF showed bigger loaf than 3H AHP SCF. So, it should be higher specific volume. Please explain more.

- Why untreated SCF showed better quality in term of textural properties and also appearance?

Reviewer 5 Report

I have reviewed the manuscript entitled “Effects of Incorporating Alkaline Hydrogen Peroxide Treated Sugarcane Fibre on The Physical Properties and Glycemic Potency of White Bread”. The novelty and significance of this work should be made more explicit, especially in the abstract and conclusion sections. There is no sufficient discussion of the results obtained. I recommended major revisions. Some comments are added below that could help authors to improve this manuscript exhaustively.

1. It is recommended that specific data be added to the Abstraction to demonstrate the improvement of in physical properties of sugarcane Fibre by alkaline hydrogen peroxide treatment.

2. Line 40-42: “Although research has proven the health effects of soluble fibres to be more pronounced than insoluble fibres, insoluble fibres such as lignocellulosic materials are more abundant and affordable.” Please provide relevant literature to support this thesis.

3. Line 51-52: “Finding optimal methods to incorporate SCF into food products, such as bread, a staple 51 food, can benefit consumers and increase the net profit of sugarcane industry.” Please provide relevant literature.

4. What are the problems with existing studies and what are the innovations of your study compared to them, please highlight.

5. How were the sample characterization methods set? please indicate the reference. Otherwise, please discuss how these conditions were chosen, as this is important for the results.

6. Please provide details on how to take samples in the texture profile analysis test. This is because a sample diameter that is too large or too small compared to the probe diameter (36R) will influence the test results.

7. The authors noted “Accordingly, the acid-insoluble lignin content of SCF decreased after the AHP treatment at 0.5 h but increased again at 3 h (Table 2).” Please provide an explanation for the increase in acid-insoluble lignin content after AHP treatment.

8. The article lacks in-depth discussion and explanation of the trends presented by the experimental results, as in line 493-494, please add.

9. Please standardize the citation format in your articles, such as Line 657.

10. Please provide higher resolution spectrograms in the manuscript.

11. Please highlight the novelty of your study in the Conclusion.

Round 2

Reviewer 1 Report

No comments

Author Response

Thank you reviewer for your comments and suggestions.

Reviewer 4 Report

Overall, the revised manuscript has been improved. However, small errors should be concerned as attached file. It can be revised during the process of corrected proof.

Author Response

Thank you for your kind suggestion. Revision has been done in line 210 and line 250 marked in red.

Reviewer 5 Report

The revised version of the manuscript Foods-2239611 has been substantially strengthened by modifications made and inaccurate expressions and unsupported conclusions have also been corrected. The novelty and significance of this work have been highlighted by revising the abstract and conclusions of the manuscript.

Author Response

Dear reviewer,

We highly appreciate your time and effort to review and offering suggestions to improve our manuscript. It was our pleasure to meet reviewer like you to review our study.

Best regards,

Authors